# The Novel AT2 Receptor Agonist β-Pro7-AngIII Exerts Cardiac and Renal Anti-Fibrotic and Anti-Inflammatory Effects in High Salt-Fed Mice

**DOI:** 10.3390/ijms232214039

**Published:** 2022-11-14

**Authors:** Yan Wang, Jonathan Yodgee, Mark Del Borgo, Iresha Spizzo, Levi Nguyen, Marie-Isabel Aguilar, Kate M. Denton, Chrishan S. Samuel, Robert E. Widdop

**Affiliations:** 1Cardiovascular Disease Program, Biomedicine Discovery Institute, Monash University, Clayton, VIC 3800, Australia; 2Departments of Pharmacology, Monash University, Clayton, VIC 3800, Australia; 3Department of Biochemistry and Molecular Biology, Monash University, Clayton, VIC 3800, Australia; 4Department of Physiology, Monash University, Clayton, VIC 3800, Australia

**Keywords:** fibrosis, AT_2_ receptor, inflammation

## Abstract

A high salt (HS) diet is associated with an increased risk for cardiovascular diseases (CVDs) and fibrosis is a key contributor to the organ dysfunction involved in CVDs. The activation of the renin angiotensin type 2 receptor (AT_2_R) has been considered as organ protective in many CVDs. However, there are limited AT_2_R-selective agonists available. Our first reported β-substituted angiotensin III peptide, β-Pro^7^-AngIII, showed high selectivity for the AT_2_R. In the current study, we examine the potential anti-fibrotic and anti-inflammatory effects of this novel AT_2_R-selective peptide on HS-induced organ damage. FVB/N mice fed with a 5% HS diet for 8 weeks developed cardiac and renal fibrosis and inflammation, which were associated with increased TGF-β1 levels in heart, kidney and plasma. Four weeks’ treatment (from weeks 5–8) with β-Pro^7^-AngIII inhibited the HS-induced cardiac and renal fibrosis and inflammation. These protective effects were accompanied by reduced local and systemic TGF-β1 as well as reduced cardiac myofibroblast differentiation. Importantly, the anti-fibrotic and anti-inflammatory effects caused by β-Pro^7^-AngIII were attenuated by the AT_2_R antagonist PD123319. These results demonstrate, for the first time, the cardio- and reno-protective roles of the AT_2_R-selective β-Pro^7^-AngIII, highlighting it as an important therapeutic that can target the AT_2_R to treat end-organ damage.

## 1. Introduction

Cardiovascular diseases (CVDs) remain the leading cause of mortality and morbidity globally, which are often initiated by cardiac or renal insult and injury, that activates the innate immune system to trigger inflammatory responses for tissue repair [1,2]. Repetitive injury leads to dysregulated extracellular matrix (ECM) remodelling, including increased tissue (TGF)-β1-mediated myofibroblast differentiation and reduced collagen degradation [3,4,5,6], which all contribute to accumulated collagen in organs (e.g., the heart and kidneys), driving towards progressive cardiac and renal fibrosis, which eventually cause organ dysfunction [7].

Current frontline therapies for cardiovascular diseases are largely dependent on the blockade of the renin angiotensin system, including angiotensin type 1 receptor (AT_1_R) blockers and angiotensin-converting enzyme inhibitors (ACEi). However, their abilities to reduce organ damage are limited [8,9], highlighting the urgency of developing novel therapeutic treatments for end-organ damage, including fibrosis.

The angiotensin type 2 receptor (AT_2_R) is upregulated following cardiovascular injury in rodents [10,11,12,13,14,15,16] and well as in human cardiac biopsies [17,18,19,20]. The stimulation of the AT_2_R has been shown to be cardio- and reno-protective in various animal models in the past decade [20,21,22]. However, there are limited AT_2_R-selective ligands that are currently available. To address this issue, our group has synthesized a library of AT_2_R-selective agonists based on the peptide sequence of AngIII peptide, which has a slightly higher affinity for the AT_2_R over the AT_1_R [23], using a β-amino acid substitution strategy [24]. We have recently reported that a single β-substitution to the amino acid proline in position 7 of Ang III (β-Pro^7^-AngIII) resulted in a more than 20,000-fold AT_2_R/AT_1_R selectivity, and this peptide evoked vasorelaxation in vitro and also reduced blood pressure in conscious spontaneous hypertensive rats (SHRs), which were blocked by the AT_2_R antagonist PD123319 [25]. More recently, we reported that β-Pro^7^-AngIII improved renal function by promoting renal vasodilation and natriuresis in normotensive rats, which were also blocked by PD123319 [26].

To date, the chronic effects of β-Pro^7^-AngIII have not been examined. On the basis of previous studies [21], we hypothesized that this peptide would exhibit anti-inflammatory and anti-fibrotic effects. To this end, we now report the effects of the novel highly selective AT_2_R agonist, β-Pro^7^-AngIII, on cardiac and renal inflammation and fibrosis following a high salt (HS) diet.

## 2. Results

### 2.1. Blood Pressure and Body Weight

After 8 weeks on a HS diet, the systolic blood pressure (SBP) was similar to that of NS-fed mice, and there was no significant effect of β-Pro^7^-AngIII on SBP (Appendix A). Cardiac hypertrophy, measured by the left ventricular weight (VW) to body weight (BW) ratio, was similar between groups: 3.74 ± 0.16 (NS); 4.10 ± 0.14 (HS); 4.00 ± 0.05 (HS+ β-Pro^7^-AngIII); 4.02 ± 0.16 (HS+ β-Pro^7^-AngIII+PD123319).

### 2.2. Cardiac and Renal Fibrosis

Cardiac and renal fibrosis were measured using picrosirius red staining to estimate collagen I and III [27] as well as by hydroxyproline for total collagen concentration [28]. HS increased cardiac and renal interstitial fibrosis by ~1.5 fold using both picrosirius red staining and hydroxyproline measurements (Figure 1 and Figure 2). This HS-induced cardiac and renal fibrosis was inhibited by the novel AT_2_R-selective peptide, β-Pro^7^-AngIII (Figure 1 and Figure 2). Importantly, the anti-fibrotic effects caused by β-Pro^7^-AngIII were abolished by the AT_2_R antagonist PD123319 using both fibrotic measurements (Figure 1 and Figure 2).

TGF-β1 is considered to be the major pro-fibrotic mediator for collagen synthesis and fibrosis development [3,4]. The HS diet increased tissue TGF-β1 levels, measured by immunofluorescence, in both the heart and kidney, which was attenuated by β-Pro^7^-AngIII whereas the co-administration of PD123319 inhibited the effects of β-Pro^7^-AngIII (Figure 3a,c and Figure 4a,b). Moreover, circulating TGF-β1 levels were significantly inhibited by β-Pro^7^-AngIII compared with the untreated HS group (Figure 4c).

TGF-β1 promotes the differentiation of fibroblasts into myofibroblasts [5,6], which are the major cellular driver for collagen production [29,30,31]. The expression of α-smooth muscle actin (α-SMA) identifies the differentiated myofibroblasts [32,33]. Cardiac α-SMA immunofluorescence was elevated by a 2-fold increase by the HS diet, which was significantly inhibited by β-Pro^7^-AngIII in a PD123319-sensitive manner (Figure 3b,d).

### 2.3. Cardiac and Renal Inflammation

Cardiac and renal phosphorylated (p)-IқBα was measured by immunofluorescence as a marker for NF-қB activity. In both the heart and kidney, HS significantly increased p-IқBα by ~1.5 fold (Figure 5a,c and Figure 6a,c). The HS-induced p-IқBα was attenuated by β-Pro^7^-AngIII in the heart and kidney, and these anti-inflammatory effects were inhibited by PD123319 (Figure 5a,c and Figure 6a,c).

The macrophage marker F4/80 was readily detected in the kidney but not the heart; the monocyte chemoattractant protein (MCP)-1 was therefore measured in the heart. HS increased renal F4/80 immunofluorescence ~2.5-fold (Figure 6b,d), while MCP-1 was modestly increased by HS in the heart (Figure 5b,d). Nevertheless, β-Pro^7^-AngIII significantly reduced cardiac MCP-1 and renal F4/80 levels whereas PD123319 inhibited the effects of β-Pro^7^-AngIII in these tissues (Figure 5d,b and Figure 6b,d).

## 3. Discussion

We have previously reported that β-Pro^7^-AngIII is a highly selective AT_2_R agonist that acutely evokes vasodilation and natriuresis [25,26]. However, the chronic effects of this AT_2_R agonist have not previously been described. In the present study, we report for the first time that β-Pro^7^-AngIII is anti-fibrotic and anti-inflammatory against HS-induced organ damage. These protective effects following AT_2_R activation were associated with reduced myofibroblast differentiation and TGF-β1 levels. Of note, this AT_2_R-selective peptide, at the dose that evoked anti-fibrotic and anti-inflammatory effects, did not affect blood pressure, indicating that the activation of AT_2_R directly contributes to cardio- and reno-protective effects, independent of systemic blood pressure.

Chronic HS intake has long been recognized as a major contributor to the development of CVDs globally [34,35]. A chronic HS diet induced cardiac and renal fibrosis in both hypertensive and normotensive animals [36,37,38,39]. Ferreira et al. reported that a HS diet (4%) significantly increased cardiac interstitial fibrosis in WKYs that was independent of blood pressure [36]. Similarly, we found that a HS diet (5%) increased cardiac and renal interstitial fibrosis in FVB/N mice without elevating blood pressure [40]. A HS diet increased cardiac and renal inflammation as well as fibrosis within 4–5 weeks [40,41,42]. In the current study, β-Pro^7^-AngIII was administered after the mice were already on a HS diet (5%) for 4 weeks; therefore, the ability of the novel AT_2_R-selective ligand to reverse established fibrosis was investigated, although we did not measure fibrosis directly after 4 weeks of HS diet in the current study. Impressively, β-Pro^7^-AngIII appeared to reverse the HS-induced fibrosis back to NS levels in both the heart and kidney, whereas these anti-fibrotic effects were blocked by the AT_2_R antagonist, PD123319, confirming that the anti-fibrotic effects induced by β-Pro^7^-AngIII were via AT_2_R activation. Consistent with the current findings, Patel et al. reported that chronic C21 treatment reduced HS-induced renal fibrosis in obese Zucker rats [43]. We have recently reported that the AT_2_R-selective peptide CGP4211 also reversed HS-induced fibrosis in the same model [40]. Therefore, our current findings are consistent with other evidence of AT_2_R-mediated anti-fibrotic effects in other disease models [21,22], including reduced cardiac fibrosis following MI injury [44,45] and in SHR-SP [46] and reduced renal fibrosis in diabetic rodents [47,48] and SHR-SP [49].

TGF-β1-induced myofibroblast differentiation induces excessive collagen production and results in organ fibrosis [3,4,5,6]. As expected, we found that HS-induced organ fibrosis was closely related to increased TGF-β1 levels in the heart and kidney. Indeed, this increased myofibroblast differentiation in the heart would have contributed to the HS-induced cardiac fibrosis. While TGF-β1 activates Smad complexes that translocate across the nuclear membrane to regulate collagen gene expression directly [50], TGF-β1 also activates many other Smad-independent, pro-fibrotic pathways such as the extracellular signal-regulated kinase (ERK), c-Jun-N-terminal kinase (JNK) and p38 mitogen-activated protein kinase pathways [50] that may contribute to pathophysiological effects. In any case, consistent with the anti-fibrotic effects, β-Pro^7^-AngIII inhibited HS-induced cardiac and renal TGF-β1, as well as circulating TGF-β1 levels, which agrees well with the previous studies showing that AT_2_R stimulation inhibits TGF-β1 levels [40,44,47,51,52].

Chronic inflammation has been considered to be the primary trigger for fibrosis development [1,2]. HS-induced organ inflammation has been reported in many hypertensive animal models [53,54,55]. In the current study, we found that HS increased cardiac and renal inflammation (increased NF-κB activity and macrophage infiltration) in normotensive mice. Under these conditions, β-Pro^7^-AngIII reduced cardiac and renal inflammation, which was entirely consistent with an AT_2_R-mediated effect, as reported using other AT_2_R-selective agonists such as C21 and CGP42212 in other disease models [48,49,51,56,57,58]. The fact that PD123319 significantly inhibited the anti-fibrotic and anti-inflammatory effects caused by β-Pro^7^-AngIII in the majority of cases, or resulted in the combination treatment being not different from HS alone, confirmed the involvement of AT_2_R in the current study. The use of PD123319 to confirm an AT_2_R-mediated mechanism was necessary, particularly since a number of previous studies investigating the chronic effects of C21 did not include PD123319 [21].

In conclusion, we demonstrated that the novel AT_2_R-selective agonist β-Pro^7^-AngIII inhibited HS-induced cardiac and renal fibrosis and inflammation in a PD123319-sensitive manner. Collectively, these findings have established that the acute AT_2_R-mediated effects of this peptide [25] have translated into cardiac and renal anti-fibrotic and anti-inflammatory effects following chronic treatment with β-Pro^7^-AngIII. These findings further highlight the AT_2_R as a novel therapeutic target for the treatment of cardiac and renal diseases.

## 4. Materials and Methods

### 4.1. Materials

β-substituted angiotensin peptide synthesis, including β-Pro^7^-AngIII, had been described in detail previously [25,59]. PD123319 was purchased from Sigma-Aldrich (St. Louis, MO, USA).

### 4.2. Animals and Treatments

Male *FVB/N* mice aged from 10 to 12 weeks were obtained from the Monash Animal Research Precinct (Clayton, Victoria, Australia) and were housed in the Department of Pharmacology animal holding facility. All experimental procedures outlined below were approved by the Monash University Animal Ethics Committee (under MARP/2013/118). Mice were housed in standard mouse cages at 21 ± 3 °C, with a 12 h light/dark cycle with free access to food and water. Mice were fed with a HS diet (5% NaCl; SF05-038 diet containing 5% NaCl; Specialty Feeds Western Australia) for 4 weeks before drug administration. Thereafter, mice were randomized into treatment groups for another 4 weeks while on a HS diet and treatments were administered via mini osmotic pumps implanted subcutaneously. The randomized groups were: HS + saline, HS + β-Pro^7^-AngIII (0.1 mg/kg/day), or HS+ β-Pro^7^-AngIII (0.1 mg/kg/day) + PD123319 (1 mg/kg/day) from weeks 5–8. Another group of mice were fed with a normal salt (NS; 0.5% NaCl) diet for 8 weeks. All treatment groups were randomly allocated and investigators were blinded to treatments and subsequent analysis. The doses were chosen based on a similar dose of β-Pro^7^-AngIII shown acutely to evoke vasodepressor effects in vivo [25].

### 4.3. Blood Pressure and Body Weight Measurements

Systolic blood pressure (SBP) was measured every 2 weeks, starting from week 0 (prior to HS diet) until week 8, using tail cuff plethysmography (MC4000 BP Analysis Systems; Hatteras Instruments Inc., Grantsboro, NC, USA). At least three cycles (with five measurements in each cycle) were performed and averaged for each animal at each time point. Body weight was measured every 2 weeks from week 0 (prior to HS diet) until week 8.

### 4.4. Analysis of Extracellular Matrix (ECM) and Inflammation

After week 8, mice were humanely killed by isoflurane overdose, and heart, kidneys and plasma were collected for ex vivo measurements. The left ventricle (LV) was separated into three portions; apex, mid-zone and base. One section from the mid-zone of the left ventricle (LV) and left kidney were embedded in OCT and immersed in methylbutane/isopentane (Merck) for slow freezing using liquid nitrogen. Next, 5 μm sections were cut using a Cryostat (Leica; CM1860) for both heart and kidney. Cardiac interstitial collagen deposition was identified from frozen sections that were stained with 0.05% picrosirius red (Polysciences Inc., Warrington, PA, USA). At the same time, a section from the mid-zone of the left kidney was fixed with 10% formalin for 24 h before picrosirius red staining (Monash Histology Platform).

Frozen cardiac and renal sections (5 μm) were used for immunofluorescence staining. Specific markers with relative antibody concentrations are summarized in Appendix A. All images were taken under 200 × magnification and quantified from an average of 6–8 fields of view for each heart and kidney section using Image J software.

The hydroxyproline content was also measured in both heart and kidney [52,60]. Equivalent tissue portions of kidney tissue (containing cortex and medulla) or left ventricular tissue (apical region) were lyophilized to dry weight, hydrolyzed in 6 M hydrochloric acid, and the collagen concentration was then calculated according to the hydroxyproline values [61].

### 4.5. Plasma TGF-β1 Measurement by ELISA

Plasma TGF-β1 levels were measured using a mouse TGF-β1 ELISA kit (R&D Systems; MB100B) according to the manufacturer’s specifications. TGF-β1 in plasma samples were activated by 1 N HCl followed by 1.2 N NaOH/0.5 M HEPES. A final dilution factor of 90 was applied in this assay as per the manufacturer’s instructions.

### 4.6. Statistical Analysis

All results were expressed as mean ± standard error of mean (s.e.m.). All statistical analyses were performed using the Prism program (GraphPad Prism software, version 9; San Diego, CA, USA). All statistical comparisons except SBP were conducted using One-Way analysis of variance (ANOVA) followed by Tukey’s post-hoc correction test for multiple comparisons between groups. SBP analysis was performed by two-way repeated measures ANOVA which allowed for within- and between-group analysis over time. For all results, *p* < 0.05 was deemed statistically significant.

## Figures and Tables

**Figure 1 ijms-23-14039-f001:**
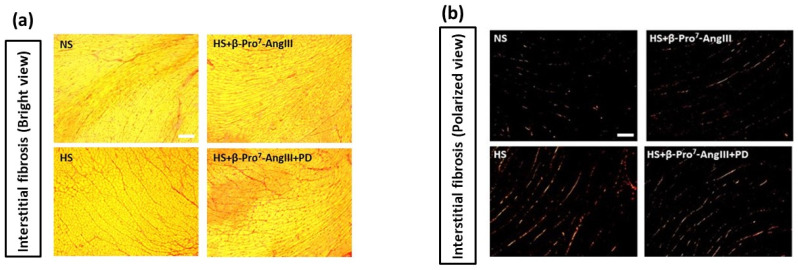
Representative (**a**) bright view and (**b**) polarized view for interstitial fibrosis sections stained with picrosirius red (indicated by red) in left ventricle (LV) obtained from male *FVB/N* mice fed on a normal salt (NS, 0.5% NaCl) diet or mice fed on a high salt (HS, 5% NaCl) diet in the presence or absence of β-Pro^7^-AngIII (0.1 mg/kg/day) +/− PD123319 (1 mg/kg/day). Scale bar for all images = 50 μm. (**c**) Group data of % area of interstitial fibrosis in LV sections. (**d**) Total tissue collagen measured by hydroxyproline assay in LV tissue from same animal groups in which picrosirius red staining was performed (n = 7–8 per group). All data are expressed as mean ± s.e.m. * *p* < 0.05, *** p* < 0.01 vs. sham; # *p* < 0.05, ## *p* < 0.01 vs. HS; & *p* < 0.05 vs. HS+ β-Pro^7^-AngIII, determined by One-Way ANOVA with Tukey correction for multiple comparisons.

**Figure 2 ijms-23-14039-f002:**
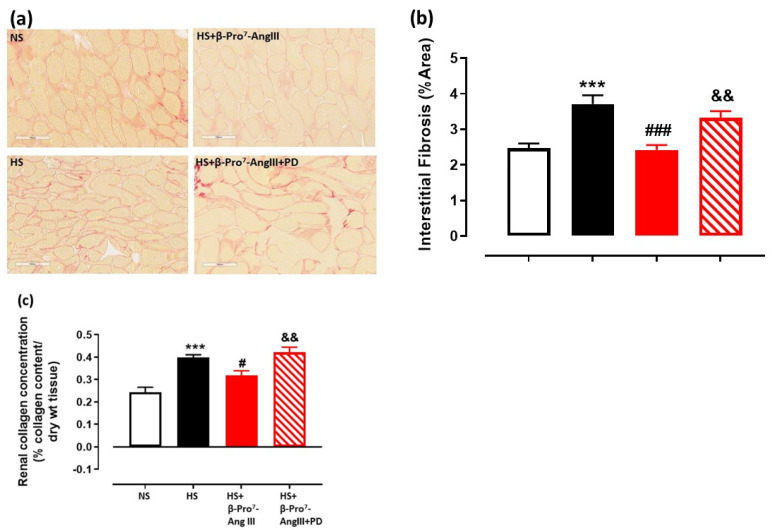
(**a**) Representative images for interstitial fibrosis stained with picrosirius red (indicated by red) in kidney sections obtained from male FVB/N mice fed on a normal salt (NS, 0.5% NaCl) diet or mice fed on a high salt (HS, 5% NaCl) diet in the presence or absence of β-Pro^7^-AngIII (0.1 mg/kg/day) +/- PD123319 (1 mg/kg/day). Scale bar for all images = 50 μm. (**b**) Group data of % area of interstitial fibrosis in kidney sections. (**c**) Total tissue collagen measured by hydroxyproline assay in kidney tissue from same animal groups in which picrosirius red staining was performed (n = 7–8 per group). All data are expressed as mean ± s.e.m. *** *p* < 0.001 vs. sham; # *p* < 0.05, ### *p* < 0.001 vs. HS; && *p* < 0.01 vs. HS+ β-Pro^7^-AngIII, determined by One-Way ANOVA with Tukey correction for multiple comparisons.

**Figure 3 ijms-23-14039-f003:**
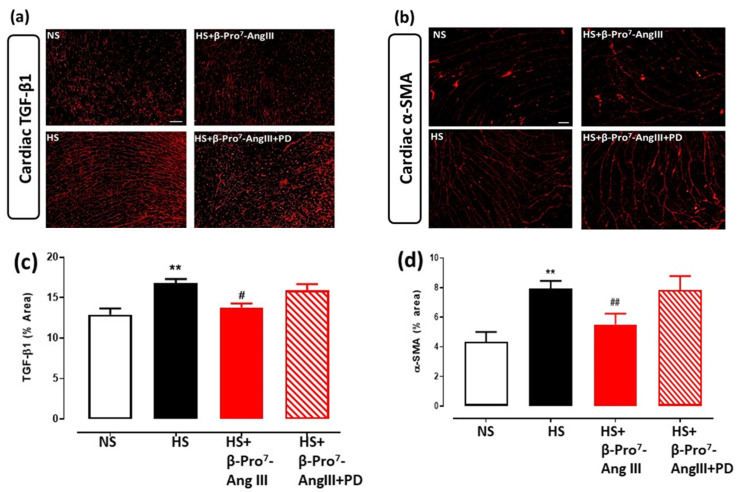
Representative images for (**a**) TGF-β1 and (**b**) α-SMA in cardiac sections obtained from male *FVB/N* mice fed on a normal salt (NS, 0.5% NaCl) diet or mice fed on a high salt (HS, 5% NaCl) diet in the presence or absence of β-Pro^7^-AngIII (0.1 mg/kg/day) +/− PD123319 (1 mg/kg/day). Scale bar for all images = 50 μm. Group data of % area of (**c**) TGF-β1 and (**d**) α-SMA in cardiac sections (n = 7–8 per group). All data are expressed as mean ± s.e.m. ** *p* < 0.01 vs. sham; # *p* < 0.05, ## *p* < 0.01 vs. HS, determined by One-Way ANOVA with Tukey correction for multiple comparisons.

**Figure 4 ijms-23-14039-f004:**
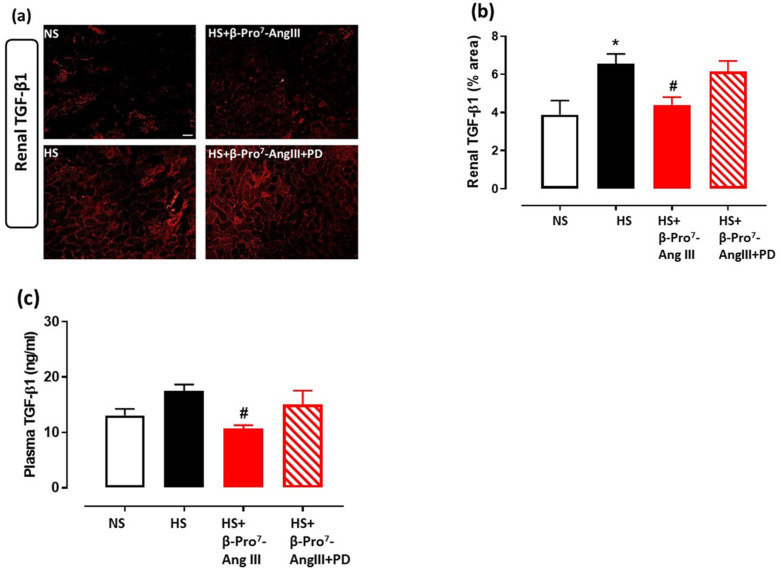
(**a**) Representative images for TGF-β1 in kidney sections obtained from *FVB/N* male mice fed on a normal salt (NS, 0.5% NaCl) diet or mice fed on a high salt (HS, 5% NaCl) diet in the presence or absence of β-Pro^7^-AngIII (0.1 mg/kg/day) +/− PD123319 (1 mg/kg/day). Scale bar for all images = 50 μm. (**b**) Group data of % area of TGF-β1 in cardiac sections. (**c**) Plasma TGF-β1 levels (n = 7–8 per group). All data are expressed as mean ± s.e.m. * *p* < 0.05 vs. sham; # *p* < 0.05 vs. HS, determined by One-Way ANOVA with Tukey correction for multiple comparisons.

**Figure 5 ijms-23-14039-f005:**
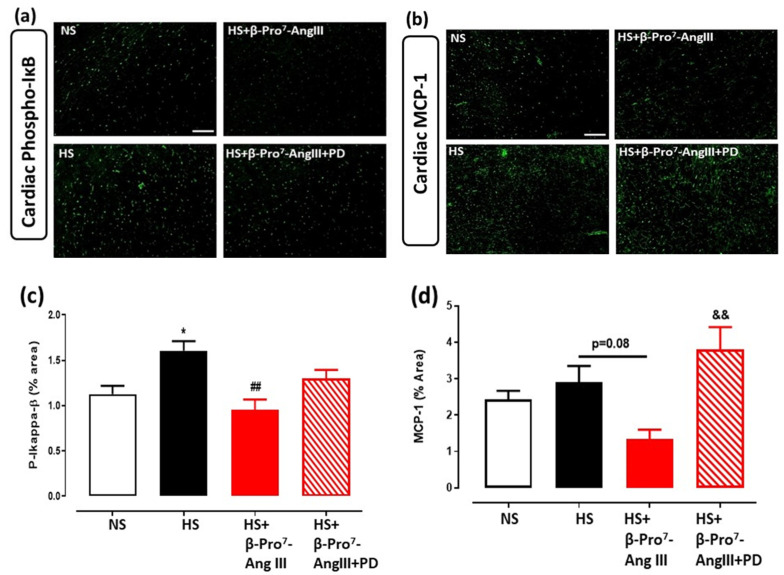
Representative images for (**a**) p-IκB and (**b**) MCP-1 in LV sections obtained from male *FVB/N* mice fed on a normal salt (NS, 0.5% NaCl) diet or mice fed on a high salt (HS, 5% NaCl) diet in the presence or absence of β-Pro^7^-AngIII (0.1 mg/kg/day) +/− PD123319 (1 mg/kg/day). Scale bar for all images = 50 μm. Group data of % area of (**c**) p-IκB and (**d**) MCP-1 in LV sections from indicated groups (n = 4–6 per group). All data are expressed as mean ± s.e.m. * *p* < 0.05 vs. sham; ## *p* < 0.01 vs. HS; && *p* < 0.01 vs. HS+ β-Pro^7^-AngIII, determined by One-Way ANOVA with Tukey correction for multiple comparisons.

**Figure 6 ijms-23-14039-f006:**
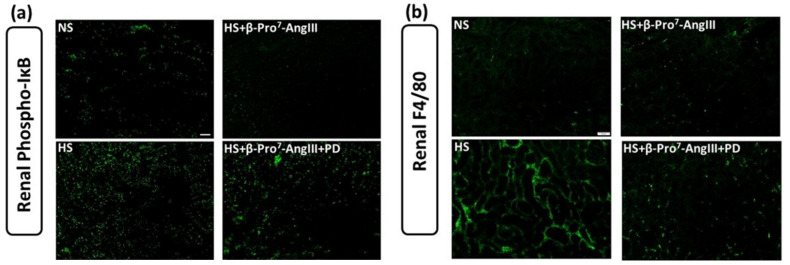
Representative images for (**a**) p-IκB and (**b**) F4/80 in kidney sections obtained from male *FVB/N* mice fed on a normal salt (NS, 0.5% NaCl) diet or mice fed on a high salt (HS, 5% NaCl) diet in the presence or absence of β-Pro^7^-AngIII (0.1 mg/kg/day) +/− PD123319 (1 mg/kg/day). Scale bar for all images = 50 μm. Group data of % area of (**c**) p-IκB and (**d**) F4/80 in kidney sections from indicated groups (n = 6–8 per group). All data are expressed as mean ± s.e.m. ** *p* < 0.01, *** *p* < 0.001 vs. sham; ## *p* < 0.01 vs. HS; & *p* < 0.05, && *p* < 0.01 vs. HS+ β-Pro^7^-AngIII, determined by One-Way ANOVA with Tukey correction for multiple comparisons.

## Data Availability

Data is available upon request. Contact Robert Widdop.

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
