# Peer review of "The Novel AT2 Receptor Agonist β-Pro7-AngIII Exerts Cardiac and Renal Anti-Fibrotic and Anti-Inflammatory Effects in High Salt-Fed Mice"

_ijms, 2022, doi:10.3390/ijms232214039_

Round 1

Reviewer 1 Report

In their manuscript entitled “The novel AT2 receptor agonist β-Pro7-AngIII exerts cardiac and 2 renal anti-fibrotic and anti-inflammatory effects in high salt-3 fed mice”, Zan Wang and co-authors report about a study, in which they examined the effect of AT2 receptor stimulation with the AT2 receptor agonist β-Pro7-AngIII, which this group has synthesized and characterized for AT2 receptor selectivity and short-term effects. This study is the first that applied β-Pro7-AngIII for chronic treatment. The authors show that in FVB/B mice fed a 5% salt diet for 8 weeks and treated with β-Pro7-AngIII from week 5 to 8 cardiac and renal fibrosis and inflammation were ameliorated coinciding with reduced TGF-beta expression and reduced cardiac myofibroblast differentiation.

This is a very coherent, well-designed study and a very well-written manuscript.

There are no major concerns.

However, importantly, the use of the term “revers” to describe the effect of β-Pro7-AngIII is not justified and should be omitted, since no data are presented on the status of fibrosis and inflammation at treatment start. Treatment may still have prevented further aggravation of fibrosis and inflammation, but not reversed existing fibrosis and inflammation.

Nevertheless, treatment start after a few weeks of HD diet is a more clinically relevant study design than treatment at start of HS diet would have been, and the authors are right in pointing this out.

Some more minor comments:

- The authors state in the discussion that basically all effects were inhibited by PD123319. However, in most cases, this inhibition was not statistically significant. This should be mentioned and discussed.

- It would be preferable if data were displayed as bar graphs which additionally show every data point.

Author Response

Thank you for your positive review of our work. Responses to your comments are listed below.

The use of ‘reverse’ to explain effects of β-Pro7-Ang III:

We accept that this phrase is less valid without a 4-week measurement before drug treatment commenced. Therefore, we have changed ‘reversed’ to ‘inhibited’ or ‘attenuated’ in many cases within the manuscript. However, in the Discussion (lines 159-163), we had already established that this treatment time is likely to represent ‘reversal’ effect. Nevertheless, we have now softened this section by indicating that we did not measured fibrosis after 4 weeks HS as well as the phrase ‘appeared to reverse’ (lines 164-165).

Effects inhibited by PD123319 were not always significant:

We would like to strongly refute the implication that PD123319 did not inhibit β-Pro7-Ang III. In 4 out of 6 figures PD123319 significantly reversed the effects of β-Pro7-Ang III. In Figs 3 and 4, while the effects of β-Pro7-Ang III plus PD123319 were not significantly different to drug alone, the combinations were also not significantly different to HS alone. Therefore, we are confident that PD123319 was exerting an inhibitory effect. Nevertheless, we have modified this point (lines 194-196) in the revised Discussion.

Histogram format:

There is no journal requirement about formatting and we would prefer to keep the figures in their current format.

Reviewer 2 Report

The paper shows the cardio- and reno-protective roles of the AT2R-selective β-Pro7 -AngIII, highlighting it as an important therapeutic that can target the AT2R to treat end-organ damage. It presents very interesting data but could have explored the effect of long-term treatment on angiotensin II and renin expression in this model. However, this does not detract from the merit of the paper. I believe it would be interesting to publish this work, as the AT2 receptor is a great pharmacological target for the future. We still do not have full knowledge of this receptor, as well as an appropriate therapeutic drug.

Author Response

Thank you for your positive review of our work. Responses to your comments are listed below.

Long term treatment on Angiotensin II and renin expression:

Thank you for this suggestion but this was beyond the scope of the current study. However, we did note in the Introduction (lines 43-44) that the angiotensin type 2 receptor (AT2R) is upregulated following cardiovascular injury in rodents and well as in human cardiac biopsies.